# Mitochondrial Respiratory Chain Supercomplexes: From Structure to Function

**DOI:** 10.3390/ijms232213880

**Published:** 2022-11-10

**Authors:** Shuting Guan, Li Zhao, Ruiyun Peng

**Affiliations:** Beijing Institute of Radiation Medicine, Beijing 100850, China

**Keywords:** mitochondria, respiratory chain, supercomplexes, assembly, cytochrome c

## Abstract

Mitochondrial oxidative phospho rylation, the center of cellular metabolism, is pivotal for the energy production in eukaryotes. Mitochondrial oxidative phosphorylation relies on the mitochondrial respiratory chain, which consists of four main enzyme complexes and two mobile electron carriers. Mitochondrial enzyme complexes also assemble into respiratory chain supercomplexes (SCs) through specific interactions. The SCs not only have respiratory functions but also improve the efficiency of electron transfer and reduce the production of reactive oxygen species (ROS). Impaired assembly of SCs is closely related to various diseases, especially neurodegenerative diseases. Therefore, SCs play important roles in improving the efficiency of the mitochondrial respiratory chain, as well as maintaining the homeostasis of cellular metabolism. Here, we review the structure, assembly, and functions of SCs, as well as the relationship between mitochondrial SCs and diseases.

## 1. Introduction

The mitochondrial respiratory chain is the basic structure for oxidative phosphorylation and plays a central role in cellular energy metabolism [1]. The mitochondrial respiratory chain consists of four enzyme complexes, including a nicotinamide adenine dinucleotide ubiquinone reductase (NADH dehydrogenase) (Complex I, CI), a succinate ubiquinone oxidoreductase (Complex II, CII), ubiquinone cytochrome oxidoreductase (Complex III, CIII), and cytochrome c oxidase (Complex IV, CIV), as well as two mobile electron carriers, ubiquinone (Co Q) and cytochrome c (Cyt c). These enzyme complexes are embedded in the inner mitochondrial membrane to form a continuous reaction system. Generally, the mitochondrial respiratory chain transfers protons and electrons from substrates to the final electronic acceptor, oxygen, and generates energy to support the phosphorylation of adenosine diphosphate (ADP) (Figure 1).

Over the past decades, models of the organization of mitochondrial respiratory chain have been controversial [2,3]. A solid model and the liquid model are two typical models for the organization of mitochondrial respiratory chain. The solid model (Figure 2A) assumes the complexes of the mitochondrial respiratory chain are anchored within a framework to ensure contact between each other with high catalytic activity [2,4]. However, with the purification of functional respiratory complexes and new understanding of the fluid nature of the mitochondrial energy transducing membrane, the liquid model (Figure 2B) has gained acceptance [5,6]. This model assumes that the components of mitochondrial respiratory chain, which participate in electron transfer, are distributed independently and are freely diffusible. Electron transfer is a multi-collisional, hindered, and long-distant diffusion process [2,7]. Compared to the solid model, the liquid model has been widely accepted.

In 2000, Schägger et al. [8] found mitochondrial supercomplexes (SCs)for the first time by blue native-page gel electrophoresis (BN-PAGE), and the enzyme activity of SCs was also confirmed. SCs are functional quaternary structures consisting of several free complexes both in *prokaryotes* and *eukaryotes* [9,10]. Compared with a free complex, SCs increase the efficiency of electron transport, reduce the production of reactive oxygen species (ROS), and stabilize the structure of free complexes [11,12,13,14]. Further studies suggested that free complexes and SCs are co-located in the inner mitochondrial membrane, have enzymatic activity and exert respiratory functions [15,16,17,18]. Moreover, SCs can be observed in BN-PAGE bands from different detergent-treated protein samples, which excludes the possibility that SCs were produced by electrophoretic techniques [15]. Furthermore, SCs have been observed in mitochondria from other species, such as *yeast*, plants, and vertebrates, by BN-PAGE [2]. Therefore, Rebeca Acín-Pérez et al. [15] proposed a plasticity model (Figure 2C) that is more flexible than the solid model or liquid model, with more complex processes for organization of the respiratory chain. Physiological activities suggest the critical roles of SCs in mitochondrial respiratory functions, and support the plasticity model for organization of the mitochondrial respiratory chain. Overall, available evidence suggests the mitochondrial respiratory chain is composed of individual complexes and SCs, both of which participate in electron transfer and have oxidative phosphorylation functions. However, the ratio of free complexes and SCs can be adjusted to adapt internal environment [16].

The assembly of SCs may be regulated by numerous factors, such as mitochondrial cristae morphology, phospholipids, assembly factors, and others. The failure of SC assembly can contribute to the development of various diseases. It has been reported that disorder of SC assembly occurs in various disease models, including neurodegenerative diseases, genetic diseases, heart failure, and cancer [1,19,20,21,22]. Therefore, the assembly and regulation of SCs are crucial for maintaining mitochondrial homeostasis and cellular functions. Here, we review the structure and assembly, the regulation mechanisms, and the functions of SCs, which will provide valuable clues for studying the efficiency of the mitochondrial respiratory chain, regulation mechanisms of cellular metabolism, and of organismal metabolic homeostasis.

## 2. Structure and Assembly of SCs

SCs can be divided into two categories: those containing CI and those without CI. The three SCs containing CI include CI + CIII, CI + CIII + CIV, and CI + CII + CIII + CIV, while the other two SCs without CI include CIII + CIV, and CII + CIII + CIV [15,16,23] (Table 1). However, SCs are heterogeneous and vary among different species, different tissue structures, and even different physiological states. In *yeast* cells, almost all the CIV binds to CIII_2_ to form SCs, and the available CIV determines the number of SCs, while in mammals, almost all the CI is bound in SCs, among which 54% is CICIII_2_CIV and 17% is CICIII_2_ [8,24].

There are two hypotheses concerning the assembly of SCs. One model assumes that the SCs are assembled after the individual complexes are formed [15], and another model assumes that the assembly of SCs starts when intermediates of individual complexes are synthesized, followed by successive assembly of the remaining subunits [13,28]. Margherita Protasoni et al. [13] found the accumulation of incompletely assembled CIII_2_ in a cell line with an MT-CyB deletion mutation. The partial subunit of CIV was found in this incompletely assembled CIII_2_. These data provided an experimental basis for the “cooperative-assembly model” for assembly as shown in Figure 3. In this model, CIII_2_, which acts as the central entity for SCs, promotes the sequential assembly of the pre-CI and CIV structural subunit modules with CIII_2_ [13,28]’ leading to a completely assembled CICIII_2_CIV. Dynamic assembly of SCs allows the cell to adapt to the internal environment and different carbon sources [10]. In *yeast*, lactate can induce the biosynthesis of CIV, which, in turn, increase the assembly of CIII_2_CIV_1-2_ and enhances OXPHOS [8]. In mammalian cell lines, increased galactose promotes the assembly of SCs to ensure that the ETC works at maximum capacity [10]. Using BN-PAGE, Chiara Greggio et al. [29] found that after 4 months exercise training, the number of CI and the assembly of SCs increased in 26 healthy, sedentary older adults.

### 2.1. CICIII_2_CIV

CICIII_2_CIV consists of a CI, a CIII dimer, and a CIV. The SC is also known as respirasome because it has all the components required to transmit electrons from NADH to oxygen [8,15,30]. *Saccharomyces cerevisiae* lacks CI, while CIV is almost absent in plants. Therefore, the respirasome is mainly detected in mammalian mitochondria. However, the number of mitochondrial respirasome varies among different species in mammals [31].

In 2016, Jinke Gu et al. [25,32] reported the structure of 1.7 mDa CICIII_2_IV in porcine heart using cryo-electron microscopy at resolutions up to 5.4 Å and 4.0 Å. The CIII dimer and CIV bind on the membrane arm of the L-shaped CI to form a transmembrane disk [25]. Interactions between the three complexes mainly occur at positions close to the membrane surface, and the interaction between CI and CIII is most obvious [33]. The subunits involved in the interaction between CI and CIII include NDUFB9 (NADH: ubiquinone oxidoreductase subunit B9), NDUFA11 (NADH: ubiquinone oxidoreductase subunit B11), NDUFB8 (NADH: ubiquinone oxidoreductase subunit B8) and NDUFB4 (NADH: ubiquinone oxidoreductase subunit B4) of CI, as well as UQCRC1 (ubiquinol-cytochrome c reductase core protein 1) of CIII [12,25,32]. CI directly interacts with CIII through the NDUFA11 subunit. In addition, the UQCRC1S251-L265, a short loop of UQCRC1, inserts into the N terminus of NDUFB9 subunit to stabilize the interaction of CI with CIII [25,32]. Moreover, the ND5 (NADH dehydrogenase subunit 5), located at the distal end of CI membrane arm, interacts with COX7C (cytochrome c oxidase subunit 7C) of CIV [12,25]. Furthermore, the COX7A subunit of CIV interacts with CIII through UQCRC1 and UQCR11 (ubiquinol-cytochrome c reductase, complex III subunit XI) [12,25,32]. Studies have demonstrated that disruption of the structure of NDUFA11 subunit, or inhibition of NDUFA11 expression can affect CI stability and result in disorder of SC assembly [34,35]. Other studies have suggested that mutations in NDUFB9 abolish CI synthesis [36], disturb NAD+/NADH balance, and promote tumor metastasis [37].

### 2.2. Other Types of SCs

The structure of CI and CIII_2_ is conserved in plants, yeast, and mammals. Therefore, the assembly of CICIII_2_ is similar to that in the respirasome, in which the NDUFB9, NDUFA11, NDUFB8, and NDUFB4 of CI interact with UQCRC1 of CIII to form CICIII_2_ [26].

CIII and CIV are assembled differently in multiple types of SCs. CIII_2_CIV_1-2_ consists of two CIII and one or two CIV. Irene Vercellino et al. [24] revealed the structures of CIII_2_CIV_1-2_ in *mice* by cryo-electron microscopy. Based on the point at which CIV interacted with CIII_2_, the structure could be divided into two types, a mature unlocked class and locked class. Moreover, the locked structures could be further divided into a locked intermediate class and a locked assembled class. The locked intermediate class contains assembled CIII_2_ intermediates, while the locked assembled class forms fully assembled CIII_2_CIV_1-2_. Unlike the mature unlocked class, the CIV of locked assembled class moves laterally much closer to CIII_2_. In the conformation of the mature unlocked class, CIV is almost rotated by 180° [24]. The assembly of CIII_2_CIV_1-2_ starts by forming the locked intermediate class, then the locked assembled class, and finally forms the mature unlocked class. During the transition, supercomplex assembly factor I (SCAFI) provides an anchor or pivot point for the rotation of CIV and plays a crucial role in stabilizing the structure of SCs [24]. In *mice*, CIII_2_CIV_1-2_ is composed 41% of the mature unlocked class and 59% Iof the locked class, which includes 21% locked intermediate class and 38% locked assembled class. However, in sheep, the unlocked class reaches 93% of the total CIII_2_CIV_1-2_, and only 7% is in the locked class [24].

Due to the lack of CI in *yeast*, the SCs are mainly CII_2_CIV_1*-*2_ and protein interaction is also different than in mammals. The Cor1 subunit of CIII linked to the Cox5a subunit of CIV [27]. Furthermore, Sorbhi Rathore et al. [27] speculated that the surface of these subunits further stabilizes the interactions of CIII_2_CIV_1*-*2_ by binding to cardiolipin molecules by observing the interface densities of Cox5a, Qcr8, Qcr9, and Rip1.

## 3. Potential Functions of SCs

Numerous studies have proposed several roles of SCs, such as maintaining the integrity of free complexes, improving the efficiency of electron transfer and regulating ROS production. The functions of SCs are shown in Table 2. Each functional advantage is complementary to each other. The loss of one function affects other functional advantages, and ultimately hinders mitochondrial energy metabolism (Figure 4).

### 3.1. Maintaining the Integrity of Free Complexes

The assembly of SCs is crucial for the assembly and stability of each free complex. Margherita Protasoni et al. [13] found that the number and activity of both CIII_2_ and CI in SCs significantly decreased in CIII_2_ subunit MT-CyB-deleted human cell lines, by using BN-GAGE, proteomics and in-gel activity assays. In addition, the subunits of CI and CIV involved in the integration of free complex, were also significantly decreased, and the assembly of CI and CIV was brought to a standstill. Both in *Paracoccus denitrificans* and the CIII-deleted *mouse* cell line L929, researchers found that the assembly of SCs was reduced. Moreover, the assembly and activity of CI was decreased, and the stability of assembled CI was reduced due to susceptibility to degradation. In addition to CI, the assembly of CIVs was also inhibited [14,38]. Moreover, Rebeca Acín-Pérez et al. [38] reported that pharmacological inhibition of CIII activity did not affect CI assembly, but physical deletion of CIII resulted in reduced CI assembly. Therefore, CIII is the center for the assembly of SCs, providing a structural and functional platform for the biogenesis of the complete mitochondrial respiratory chain [13].

### 3.2. Improving the Efficiency of Electron Transfer

The assembly of SCs facilitates the diffusion of ubiquinone and cytochrome c by shortening the distance between the complexes, and by increased electron transfer efficiency.

Cristina Bianchi et al. [11] proposed a model in which the existence of electron channels between mitochondrial respiratory chain complexes, and specific ubiquinone and cytochrome c could spread and transfer electrons, and the mitochondrial respiratory chain complex could control electron flux. Although the substrate channel theory is still controversial, growing evidence suggests that ubiquinone and cytochrome c are free to diffuse between the complexes [1]. James A. Letts et al. [12] reported that the active sites of each complex opened to the inner mitochondrial membrane at an interval of 10 nm in sheep mitochondria, detected by cryo-electron microscopy at a resolution of 5.8 Å. Although no protein-mediated substrate channels connect the individual SCs, ubiquinone and cytochrome can flow freely between them for electron transfer. These structural arrangements not only shorten the diffusion of the transmitters, but also reduce electron leakage [12]. Jens Berndtsson et al. [40] found that the efficiency of electron transfer was reduced in *yeast* when the interaction between CIII and CIV was disrupted by altering the *COR1* subunit. Introducing endogenous or exogenous cytochrome c significantly increased the growth of *yeast*. Agnes Moe et al. [39] also reported that cytochrome c with a positive charge can slide along the negatively charged surface of CIII_2_CIV in yeast cells. These results suggest that SCs may shorten the diffusion distance of cytochrome c to enhance electron transfer efficiency.

Studies have demonstrated that the enzyme activities of CIII_2_ and CIV in SCs are 1.7 and 1.9 times higher than those in free complexes in mammals, resulting in stronger catalytic activity for the redox reactions of cytochrome c and higher efficiency for electron transfer [24]. Compared to free CIII and CIV, CIII_2_CIV improved efficiency of mitochondrial respiration and ATP synthesis [16].

### 3.3. Regulating ROS Production

Mitochondria-derived ROS are mainly produced by CI and CIII, which may due to decreased electron flux between complexes [41,42]. The assembly of SCs can reduce ROS production by increasing electron transfer efficiency [11,12]. 

Irene Lopez-Fabuela et al. found that ROS production in astrocytes was higher than that in neurons. Further analysis showed that mitochondrial complex I is predominantly assembled into supercomplexes in neurons, whereas abundant free complex I can be detected in astrocytes. In CI over-expressed and knockdown models, it has been demonstrated that assembly of SCs can decrease ROS production [43]. In the respirasome, UQCRFS1, the distal subunit of CIII, COX6a2, and the subunit of CIV, were tightly associated to prevent the oxidation of ubiquinone and reduce ROS production in CIII [12,33]. Moreover, disrupting the structure of SCs increased the production of ROS. Evelina Maranzana et al. [41] found that dissociating CI from SCs by dodecyl maltoside (DDM) increased ROS production. In yeast cells that lacked regulatory SCs assembly factors, elevated ROS could be detected [44].

## 4. Regulation of SCs

Many studies have investigated the structure of SCs, and have found several factors regulating SC assembly. The regulation of SCs is shown in Table 3.

### 4.1. SCAFI

The cytochrome c oxidase subunit 7A-related protein (COX7a2L), also named SCAFI, plays critical roles in regulating the assembly of CIII and CIV. However, SCAFI is only present in SCs containing CIII_2_CIV, but not in free CIII and CIV [16]. SCAFI has two phenotypes, a long type containing 113 amino acids, and a short type encoding 111 amino acids. The phenotype of SCAFI varies among species and strains. For example, the phenotype is different between *C57/6J* and *Balb/CJ mice* [16,17]. The long-type SCAFI in mammals prompts the assembly of CIII_2_ and CIV into SCs, whereas the short-type SCAFI results in deficiency of CIII_2_CIV in strains [16,23].

SCAFI, a homolog of the COX7a subunit of CIV [12], has a similar amino acid sequence to COX7a. However, SCAFI contains an extra domain that can bind to CIII [23]. *Mouse* CIII_2_CIV was analyzed by cryo-electron microscopy and shows that SCAFI forms a physical linkage between both complexes [24]. Studies have revealed that the N terminus of SCAFI interacts with the N terminus of sub9 by electrostatic effects and is embedded in the cavity of CIII_2_. SCAFI also binds to CIV via the C terminus of the transmembrane helix at the COX7a subunit. Therefore, COX7a is replaced by SCAFI during the assembly of SCs containing CIII and CIV, and SCAFI bridges CIII and CIV to form CIII_2_CIV [12,16,23,24]. SCAFI is not only an assembly factor, but also participates in the interaction between CIII and CIV to stabilize SCs [24].

Although CIII_2_CIV does not assemble in *mice* because *mice* congenitally express short-type SCAFI, a certain number of respirasomes and free complexes can be detected in the hearts of C57 mice. No severe abnormalities of respiratory chain functions have been observed in mice [2,17,23]. In a SCAFI-deficient *zebrafish* model, the interaction between CIII and CIV was disrupted, which caused a significant reduction in CIII_2_CIV, as well as respirasomes. These changes resulted in a substantial impairment of respiratory functions, and finally led to a significant reduction in body size and fertility [45]. It is assumed that congenital deletion of SCAFI in mice regulates the type and ratio of SCs to compensate for the deficiency of CIII_2_CIV. However, gene silencing of SCAFI may cause serious disorder of mitochondrial respiratory functions due to limited compensatory capacity. Further analysis has shown that SCAFI was absent in both CICIII_2_ and the respirasome, suggesting that SCAFI is an CIII_2_CIV specific assembly factor [24].

### 4.2. Respiratory Supercomplex Factor 1

Respiratory supercomplex factor 1 (Rcf1) and respiratory supercomplex factor 2 (Rcf2) belong to the hypoxia inducible gene domain (HIGD) family. Rcf2 is expressed in yeast cells, while Rcf1 and its homologs are expressed in the plants and mammals.

In yeast cells, both Rcf1 and Rcf2 are subunits that made up the CIV and SCs. Rcf1 is necessary to maintain cellular respiration, and affects the assembly of CIV and SCs by regulating cytochrome c oxidase subunit VIa (COX13) [46]. Moreover, Rcf1 also directly interacts with CIII and CIV to form CIII_2_CIV [44]. Rcf1 knockout results in a significant decrease of CIII_2_CIV and an obvious increase of free CIII_2_ and CIV, which increases ROS production and elevates oxidative stress in *yeast* cells [44,46]. In an Rcf1-overexpressed model, both CIII_2_CIV and CIII_2_CIV_2_ are significantly increased while free CIV is absent, suggesting that elevated Rcf1 expression can promote the formation of SCs [44].

HIGD1A and HIGD2A, two homologs of Rcf1 in mammals [49], are two important subunits of CIV that play similar roles as Rcf1 [44,46,50]. It was initially reported that HIG1A could be strongly induced by hypoxia in an HIF-1-dependent manner. Although HIGD1A is not essential for the basic functions of mitochondria in mammals, it is critical for enhancement of electron transport and improving respiratory efficiency. Studies also showed that HIG2A knockdown led to depletion of CIV-containing SCs, especially CICIICIIICIV, CICIII_2_CIV, and CIII_2_CIV_1-2_ [44,46]

### 4.3. Other Regulatory Factors

In addition to the above-mentioned factors that regulate the assembly of SCs, other regulatory networks containing phospholipids and cytochrome c, have also been identified. Several studies found large amounts of phospholipids, such as cardiolipin (CL), phosphatidylethanolamine (PC), and phosphatidylcholine (PE), in mitochondrial SCs [30,47,51]. Cryo-electron microscopy with high-resolution has revealed that lipid molecules affect the enzyme complex catalytic activity by participating in interactions between complexes in SCs. For example, CL, PC, and PE interact with NDUFA11 [32,51,52]. Moreover, CL is present in the inner mitochondrial membrane and plays important roles in stabilizing the structure of SCs [53,54,55]. Knockdown of the *Crd1* gene that encodes cardiolipin synthase, impairs CL synthesis, reduces assembly of CIII_2_CIV_2_ complexes, and increases free CIII_2_ and CIV_2_ [47]. Moreover, a small number of assembled SCs are not as stable as those in wild-type cell lines [9,30].

Cytochrome c is not only a transmitter of electrons in the mitochondrial respiratory chain, but a pivotal molecule in the apoptosis-associated signaling pathway. It is well-known that cytochrome c is required for CIV assembly [48]. Uma D. Vempati et al. [53] found that knockdown of cytochrome c gene completely abolished CIV activity, decreased CIII activity to 50% of the normal level, and significantly reduced the activity of cardiolipin synthase, which in turn affected assembly of SCs.

## 5. SCs in the Development of Diseases

Disorder of the assembly of SCs, due to gene mutation or environmental stimulation, had been widely reported in various diseases. The decrease or absence of SCs has emerged as a possible secondary cause of these diseases (Table 4). Failure of SC assembly might result in weak adaptability to environmental changes, due to impaired mitochondrial energy metabolism.

Alzheimer’s and Parkinson’s disease are two major neurodegenerative diseases. Obvious amyloid deposition, mitochondrial dysfunction, and oxidative stress can be observed in brain tissue of AD and PD patients [21]. Amyloid accumulation alters phospholipids of the mitochondrial inner membrane, such as increasing ceramide, which results in dysfunction of mitochondrial energy metabolism by affecting assembly and function of SCs [22]. Studies have also found that the SCs are lower in the cerebral cortex of aging rats than that in young rats [57]. It had been demonstrated that functional deficiencies of *PINK1* and *DJ-1* are closely associated with PD. *PINK1* knockout impaired mitochondrial division, decreased the enzymatic activity of CI and CIV, as well as reduced the assembly of SCs [21,58], while deletion of *DJ-1* in dopaminergic neurons decreased the assembly and activity of CI in mitochondria [21].

Barth syndrome is an X-linked recessive inheritable disease mainly caused by mutations in the Tafazzin (TAZ) gene, which leads to disorder of cardiolipin remodeling [59]. In mouse model of Barth syndrome, Tafazzin knockout by doxycycline significantly impaired cardiolipin metabolism, reduced SCs and increased free CI. Moreover, it was reported that Tafazzin knockout resulted in disruption of mitochondrial structure, and mitochondrial strings and vesicular mitochondrial cristae were observed [19]. Leigh syndrome (LS), also known as subacute necrotizing encephalomyelopathy (SNEM), is a rare mitochondrial disease caused by a genetic deficiency [60]. Fabian Baertling et al. [20] found impaired CI synthesis in fibroblasts from *NDUFAF4*-mutated Leigh syndrome patients that decreased the assembly of CICIII_2_ and CICIII_2_CIV.

## 6. Conclusions and Future Perspectives

With the development of various advanced technologies, such as BN-PAGE, 2D BN-PAGE and cryo-electron microscopy, the location of SCs in the inner mitochondrial membrane has been demonstrated, and researchers have focused on exploring the structure and assembly of SCs [15,25,32,33]. The plasticity model of respiratory chain has been attracting more and more attention [2,3]. However, more experimental evidence is required to support the plasticity model, and the processes of assembly should be further investigated. There are several limitations to study of the assembly of SCs. First, the detergents used in isolation and purification of SCs inevitably impact the components of SCs. Therefore, isolated SCs are always different from intact SCs. Second, the varied methods of sample preparation used for cryo-electron microscopy techniques may affect results [21]. Therefore, it is necessary to optimize current technologies or develop novel methods for uncovering the assembly and functions of SCs.

Moreover, the functions of SCs, such as reducing ROS production and increasing electron transfer efficiency through substrate channels, still need to be verified in further in vitro and in vivo models [1]. It also has been widely demonstrated that abnormalities of SCs participate in the pathological processes of various diseases, especially neurodegenerative diseases, and impaired assembly of SCs has been emerged as a major cause. However, the potential mechanisms still need to be further investigated.

## Figures and Tables

**Figure 1 ijms-23-13880-f001:**
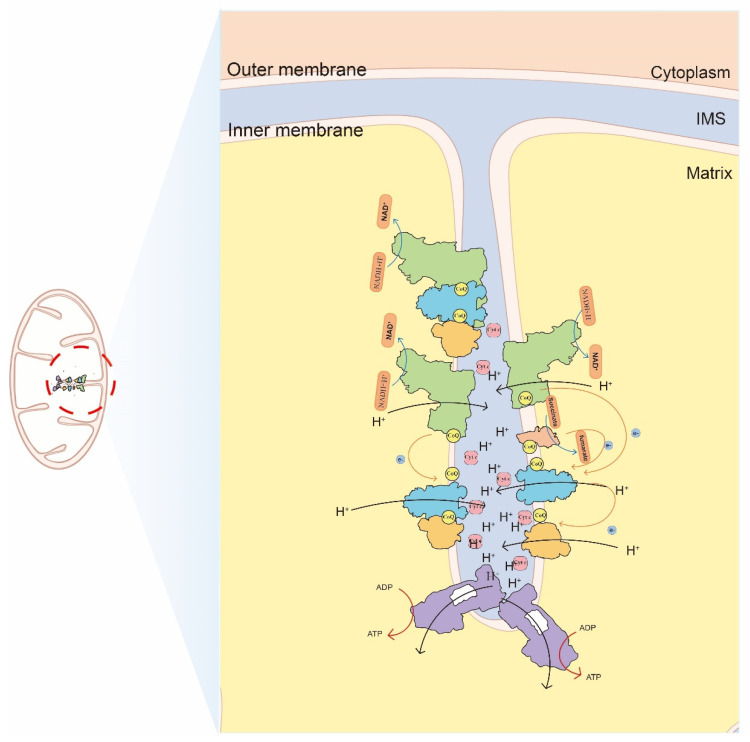
Mitochondrial cristae and the oxidative phosphorylation located on cristae. The free complexes and SCs transport electron to others though two mobile electron carriers and generate ATP. The position of cytoplasm, matrix, intermembrane space (IMS), cristae and inner membrane are indicated. The complexes are distinguished by different colors: CI: green, CII: pink, CIII dimer: blue, CIV: orange, CV: purple.

**Figure 2 ijms-23-13880-f002:**
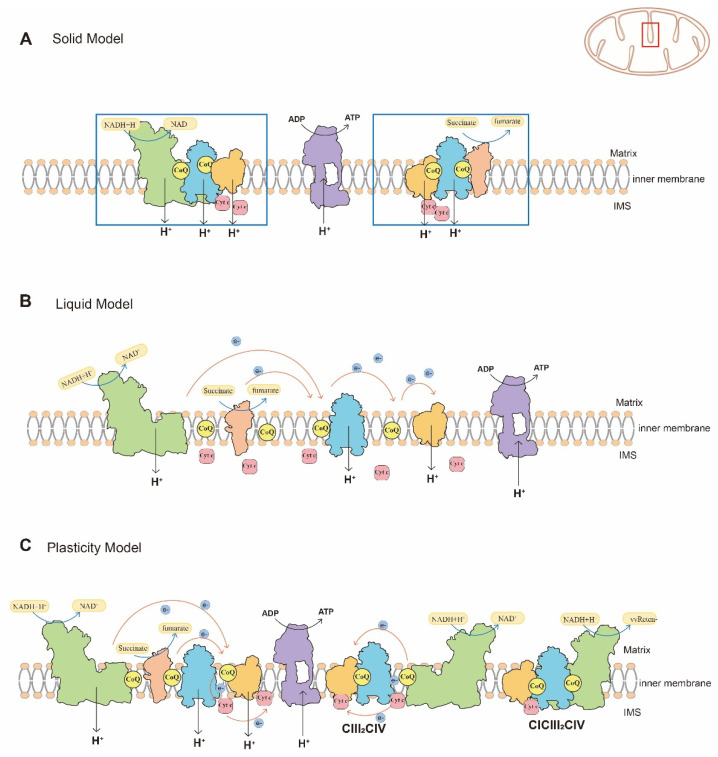
Schematic representation of the different assembly models of the mitochondrial respiratory chain. (**A**) The solid model of mitochondrial respiratory chain. The mitochondrial respiratory chain complexes and electron carriers are anchored within a framework and interact with others tightly to ensure high catalytic activity. (**B**) The liquid model of mitochondrial respiratory chain. Respiratory chain complexes are freely located on the inner mitochondrial membrane and are connected by freely diffusible electron carriers. (**C**) The plasticity model of the mitochondrial respiratory chain. In this model, individual complexes and SCs participate in electron transfer collectively or individually to adapt to the internal environment. The position of cytoplasm, matrix, the intermembrane space (IMS), cristae and the inner membrane are indicated. Green = CI, pink = CII, blue = CIII dimer, orange = CIV, purple = CV.

**Figure 3 ijms-23-13880-f003:**
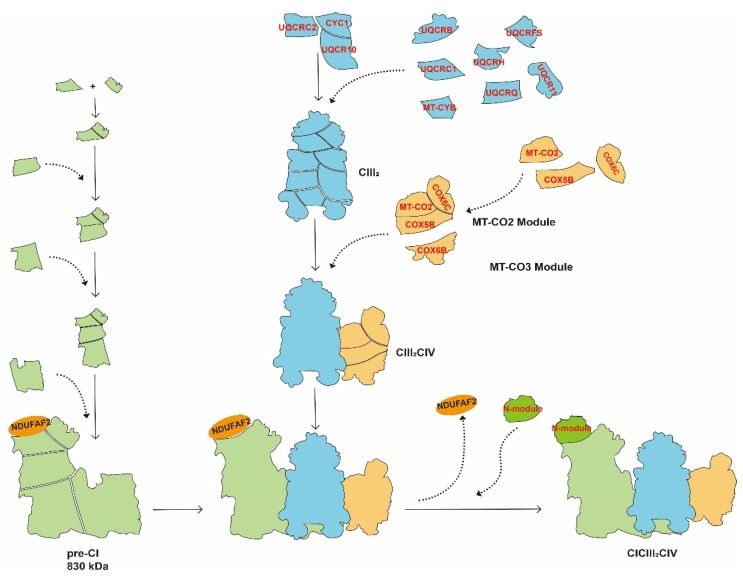
Assembly of CICIII_2_CIV by the cooperative-assembly model. CYC1, QUCRC2, and UQCRC10 are assembled to the pre-CIII_2_ stage and then other subunits are assembled successively to generate CIII_2_, which acts as the central entity for SCs. At the early stage of assembly of CIII_2_, CI builds up as a pre-CI of ~830 kDa in which the binding site of the N-module subunit is occupied by NDUFA12 to stabilize the pre-CI. In addition, several CIV subunits belonging to the MT-CO2 module (MT-CO2, COX5B, and COX6C) and the MT-CO3 module (COX6B1) interact with CYC1 and UQCR10. Then, CIII_2_ binds with pre-CI. Finally, NDUFAF2 is released from pre-CI, and the N-module is assembled to form the complete CICIII_2_CIV. The SCs are distinguished by different colors: CICIII_2_CIV: green, blue, and orange; CIII_2_CIV: blue and orange; CIII_2_: blue.

**Figure 4 ijms-23-13880-f004:**
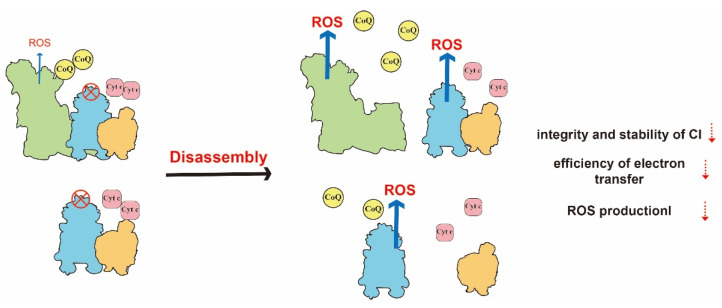
Illustration of the dysfunction of the respiratory chain induced by the disassembly of SCs. With the defect of CIII as an example, the failed assembly of SCs results in a longer diffusion distance decreased efficiency of electron transfer, and increased ROS production. In addition, the stability of CI is reduced.

**Table 1 ijms-23-13880-t001:** Structure and important subunits of various SCs in different models.

Type	Structure	Important Subunit	Model	Ref.
CI-CIII-CIV	CICIII_2_CIV	NDUFB9, NDUFA11, NDUFB8, NDUFB4, ND5, UQCRC1, UQCR11, COX7A, COX7C.	*Porcine*	[25]
	CICIII_2_CIV	NDUFB2, NDUFB3, NDUFB4, NDUFB7, NDUFB9, NDUFB10, NDUFB11, ND5; UQCRQ, UQCRC1, UQCRH, UQCR11, UQCRFS1, COX5b, COX7a, COX7c, COX8b.	*Ovine*	[12]
CI-CIII	CICIII_2_	NDUFB9, NDUFA11, UQCR10, UQCRQ, UQCRC1.	Mammalian	[26]
CIII-CIV	CIII_2_CIV	UQCRFS1, Sub7, Sub10, COX3, COX5B, COX6A2.	Mammalian (*mouse* and *ovine*)	[24]
	CIII_2_CIV_1-2_	Cor1, Cox5a.	*Yeast*	[27]
CI-CII-CIII-CIV	NA	NA	*Mouse*	[15]
CII-CIII-CIV	NA	NA	*Mouse*	[15]

**Table 2 ijms-23-13880-t002:** Functions of SCs in various models.

Function	SCs	Model	Ref.
Maintaining the integrity of free complexes	CICIII_2_CIV	MT-CYB-deficient *human* cell line	[13]
	CICIII_2_CIV	*Mouse* L929 cells	[38]
	CICIII_2_CIV	*Paracoccus denitrificans*	[14]
Improving the efficiency of electron transfer	CIII_2_CIV	Mammalian (*mouse*&*ovine*)	[24]
	CICIII_2_CIV; CICIII_2_; CIII_2_CIV	*Mouse*	[16]
	CIII_2_CIV	*Yeast*	[39]
	CICIII_2_CIV	*Ovine*	[12]
	CIII_2_CIV_1-2_	*Yeast*	[40]
Regulating ROS production	CICIII_2_CIV	*Ovine*	[12]
	CICIII_2_CIV	*Bovine*	[33]
	CICIII_2_	*Bovine*	[41]

**Table 3 ijms-23-13880-t003:** Regulatory factors at different regulatory site of SCs in diverse models.

Regulatory Factors	Regulatory Site	Model	Ref.
**SCAF1**	CIII_2_CIV	*Mouse*	[16,17,23]
	CIII_2_CIV	*Ovine*	[12]
	CIII_2_CIV	*Zebrafish*	[45]
**HIGD family**			
Rcf 1	CIV	*Yeast*	[46]
HIGD2A	CIV	*Mouse* C2C12 cells	[44]
**Phospholipids**	CICIII_2_CIV	*Porcine*	[32]
**Cytochrome c**	CIII_2_CIV	*Saccharomyces Cerevisiae*	[47]
	CIV	*Saccharomyces Cerevisiae*	[48]

**Table 4 ijms-23-13880-t004:** The role of SCs in the progression of diseases.

Disease	SCs	Progress	Ref.
AD	CI–CIII–CIV; CI–CIII; CIII–CIV.	Retention of amyloid at MAM→Failure of the assembly of SCs→Dysfunction of OXPHOS	[22]
PD	CICIII_2_CIV; CICIII_2_; CIII_2_CIV.	Absent of *Dj1*→Defect in the assembly of complex I→Hind assembly of SCs	[56]
Barth Syndrome	All SCs	Absent of cardiolipin	[19]
Leigh Syndrome	CICIII_2_CIV_n_; CICIII_2_; CIII_2_CIV.	Mutation in *NDUFAF4*	[20]

## Data Availability

Not applicable.

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
