# Peer review of "Mitochondrial Respiratory Chain Supercomplexes: From Structure to Function"

_ijms, 2022, doi:10.3390/ijms232213880_

Round 1
Reviewer 1 Report
The review article will undoubtedly be of interest to readers. However, before publishing, I propose to make some corrections:
1. After the phrase «In 2000, Schägger et al. [6] firstly found mitochondrial supercomplexes (SCs) by blue 45 native-page gel electrophoresis (BN-PAGE) and the enzymatical activity of SCs was also 46 confirmed.» (lines 45-47), a brief introductory definition of what is meant by supercomplexes and how they differ from free complexes should be added.
2. The text describes 5 types of supercomplexes, but only three of them are marked in Table 1. The absence of a detailed caption for this table makes the logic of its construction incomprehensible.
3. Line 93 –« The SCs were also known as…» - I believe that «This SC is also known as…» is meant by this, otherwise the point of separating this text into a subsection with such a heading is generally incomprehensible. That is, only this complex is called respirasome?
4. Lines 120-127 - It is unclear why this text, which apparently refers to all supercomplexes, is placed in the section «2.1. CICIII2CIV». If we are talking only about CICIII2CIV, then this should be indicated in this text.
5. Lines 133-143 – It is unclear what is meant by locked and unlocked classes.
6. Table 2 - It is unclear what is meant by CIII-CIV in the left column
7. Line 173 – “Rebeca Acı´n-Pe´ rez” - should be correct
8. Lines 211-212 – “Therefore, the energy was produced by oxidative phosphorylation in neurons, while was generated by glycolysis in astrocytes [38].” The meaning of the phrase and its logic connection with the previous one are not clear; maybe some part of it is missing
9. Why is there no information about SCAFI and other regulators in table 3?
10. More detailed captions and explanations for the figures and tables are needed.
Author Response
Ph.D. Li Zhao
Institute of Radiation Medicine, Beijing 100850, China.
Email: lillyliz@163.com
September 19, 2022
Prof. Editor
International Journal of Molecular Sciences
Manuscript ID: ijms-1987250
Dear Editor,
Thank you very much for your comments on Oct.17, 2022, and the referees’ reports concerning our paper entitled “Mitochondrial Respiratory Chain Supercomplexes: From Structure to
Function” (Manuscript ID: ijms-1987250). Those comments are valuable for revising and improving our paper with important guiding significance. We have studied the comments carefully and have made corrections. The “point-by-point response” has been attached. In addition, we have double-checked the format to comply with the authors’ guidelines for International Journal of Molecular Sciences.
We expect our revised version to be acceptable for publication in International Journal of Molecular Sciences.
We look forward to hearing from you soon.
Thank you very much.
Sincerely yours,
Li Zhao
Professor of Medicine,
Bioelectromagnetic and Molecular Biology
Response to Reviewer 1,
Point 1: After the phrase «In 2000, Schägger et al. [6] firstly found mitochondrial supercomplexes (SCs) by blue 45 native-page gel electrophoresis (BN-PAGE) and the enzymatical activity of SCs was also 46 confirmed.» (lines 45-47), a brief introductory definition of what is meant by supercomplexes and how they differ from free complexes should be added.
Response 1: Thank you for your positive comments. Based on your comments, we have added the concept of SCs and the differences with free complexes as followed.
SCs are functional quaternary structures consisted of several free complexes both in prokaryotes and eukaryotes [50, 60]. Compared with free complex, SCs could increase the efficiency of electron transport, reduce the production of reactive oxygen species (ROS), and stabilize the structure of free complexes [11-14]. (line 61-64)
Point 2: The text describes 5 types of supercomplexes, but only three of them are marked in Table 1. The absence of a detailed caption for this table makes the logic of its construction incomprehensible.
Response 2: Thank you for your suggestion. We have added other two types of SCs. (line 124)
Point 3: Line 93 –« The SCs were also known as…» - I believe that «This SC is also known as…» is meant by this, otherwise the point of separating this text into a subsection with such a heading is generally incomprehensible. That is, only this complex is called respirasome?
Response 3: We are sorry for our carelessness. That’s right. Only CICIII2CIV is called respirasome. And, we have revised it in. (line 128-129)
Point 4: Lines 120-127 - It is unclear why this text, which apparently refers to all supercomplexes, is placed in the section «2.1. CICIII2CIV». If we are talking only about CICIII2CIV, then this should be indicated in this text.
Response 4: Thank you for your comments.
We have adjusted it to the second paragraph of the section ‘2. Structure and Assembly of SCs’. (line 108-115)
Point 5: Lines 133-143 – It is unclear what is meant by locked and unlocked classes.
Response 5: According to the point of CIV interacted with CIII2, the structure of CIII2CIV1-2 could be divided into two type, mature unlocked class and locked class. Briefly, there two class of CIII2CIV1-2 represents different assembly of SCs. Moreover, we have revised the description in manuscript as followed.
Irene Vercellino et al. [20] revealed the structures of CIII2CIV1-2 in mice by cryo-electron microscopy. And based on the point of CIV interacted with CIII2, the structure could be divided into two types, mature unlocked class and locked class. (line 169-172)
Point 6: Table 2 - It is unclear what is meant by CIII-CIV in the left column
Response 6: We are sorry for our carelessness. It should be revised to “Regulating ROS production”, and we have revised it in manuscript. (line 201)
Point 7: Line 173 – “Rebeca Acı´n-Pe´ rez” - should be correct
Response 7: Thank you. We have corrected it in the manuscript. (line 70 and line 217)
Point 8: Lines 211-212 – “Therefore, the energy was produced by oxidative phosphorylation in neurons, while was generated by glycolysis in astrocytes [38].” The meaning of the phrase and its logic connection with the previous one are not clear; maybe some part of it is missing
Response 8: Thank you for valuable comment. We have revised it as follows.
Irene Lopez-Fabuela et al. found that ROS production in astrocytes was higher than that in neurons. Further analysis explored that mitochondrial complex I is predominantly assembled into supercomplexes in neurons, whereas abundant free complex I could be detected in astrocytes. In CI over-expressed and knockdown models, it has been demonstrated that assembly of SCs could decrease ROS production [38]. (line 255-259)
Point 9: Why is there no information about SCAFI and other regulators in table 3?
Response 9: We have added corresponding content in Table 3. (line 273)
Point 10: More detailed captions and explanations for the figures and tables are needed.
Response 10: We have revised the captions and explanations according to your suggestions.

Reviewer 2 Report
Figure 1: This figure itself shows rather few information and takes a lot of space. It could be the first sub-figure of a complete and comprehensive illustration of the electron transport chain (ETC). As a matter of fact, in line 31, the writer referred it as “Figure 1 A”, but the illustration did not go on due to some reason.
Figure 2: Same problem as Figure 1. Actually, figure 1 and figure2 of this manuscript could be merged, but still, much more information should be provided to display the overall framework of ETC and OXPHOS.
All tables should be polished to improve their clearness and aesthetic.
Suggestions:
The solid/liquid/plasticity assembly mode of SCs is kind of general and basic and should not take more than a few sentences to elaborate in the introduction rather than several paragraphs and two figures. As the manuscript is entitled “from structure to function”, the researcher and writer should really focus on the exciting development of those two areas. As SCs are several higher-order complexes of ETC whose assembly and structures are dynamic, the writer should emphasize the assembly of SCs in different ETC stage and its possible function. To achieve these goals, sufficient illustrations with precise description should be provided.
Author Response
Dear Editor,
Thank you very much for your comments. We have studied the comments carefully and have made corrections. Since the picture and table cannot be displayed in the box, so I uploaded my point-by-point response to the attachment. Please check it. And thank you again for your suggestions on the manuscript.
Sincerely yours,
Shuting Guan

Round 2
Reviewer 2 Report
Those comments below are my personal opinions and they do not need to be answered one by one.
1. When I suggested “Figure 1 and Figure 2 could be merged”, what I expecting was their combination makes a good start point to bring the readers MORE INFORMATION, but NOT a retrospect of the assembly models, as I have said in point #4 that this classification is too simple and old-school and it need no more than a few sentences to be delivered.
2. When we talk about the “structure and function” of a super complex, we see it as a dynamic process, like the growing up of a person. Much more attention should be paid to the change of subunits and their interaction during the LIFE STAGE of the super complex. A step-by-step illustration of the complex is optimal. I understand it can not be perfectly done in the current, but the writer should really show their effort toward this destination, by providing NEW, CLEAR, INTEROPERABLE illustrations. One more thing, always use figure/illustration as your top-priority weapon to deliver your interpretations.
Author Response
Dear Editor,
Thank you for your valuable and constructive suggestions. We have added two figures according to the your suggestions, and also ajusted the serial number of figures. Since the picture and table cannot be displayed in the box, so I uploaded my response to the attachment. Please check it. And thank you again for your suggestions on the manuscript.
Sincerely yours,
Shuting Guan
